# Economic Calculus Qua an Instrument to Support Sustainable Development under Increasing Risk

**Grzegorz Drozdowski** 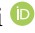

Department of Economics and Finance, Faculty of Law and Social Sciences, Jan Kochanowski University in Kielce, 25-369 Kielce, Poland; drozdowskigp@gmail.com

**Abstract:** Investment decisions in the field of sustainable development should be taken based on an economic calculation, taking into account the analysis of a diverse economic environment. The economic calculus of an enterprise is treated as a kind of way of thinking about the rationality of decisions made by an entrepreneur. In the case of sustainable development, the economic calculus serves as an instrument to support the selection of the investment measure. The result of the economic calculus is based on various types of economic parameters, which are subject to frequent changes and high risk. A risk-based financial account may be of little use in the context of the unpredictability of the forecasted situations. In the article, I attempted to determine the importance of a variable interest rate in the economic calculus of a company as an instrument to support sustainable development. For this purpose, I modified the Net Present Value (NPV) meter, which contains actual (variable) discount rates.

**Keywords:** economic calculus; sustainability; variable discount rate; economic calculus of sustainability; financial management; risk; uncertainty

## 1. Introduction

The financial management process determines the development of the economy in the period of increased risk. Companies are currently undergoing multifaceted changes that directly affect the efficiency of capital use and allocation. The resource allocation mechanism is one of the most critical factors determining the level of social sustainability. The dynamics of economic growth, as a measure of social welfare growth, is determined by the group and structure of investments (Dobrowolski and Sułkowski 2020). It is an undeniable fact that without a permanent increase in the volume of investment, it is impossible to achieve a high level of society's wealth (Anand and Sen 2000; Ionescu et al. 2018; Panait et al. 2014).

The resulting pandemic crisis and its impact on company management are characterized by the occurrence of many factors disrupting development processes. In this type of situation, there is an overlap between two groups of crisis factors (Dobrowolski 2020). First of all, companies are subject to constant adaptation as part of the ongoing changes in their internal functioning. Secondly, enterprises, being part of the external system of crisis impact, must take into account dynamic adaptation behaviors, whose effectiveness depends mostly on the value of development investments made.

The increasing level of risk, and especially uncertainty caused by the COVID-19 pandemic, contributes, on the one hand, to faster implementation of the microeconomic development mechanism, based on the paradigms of economic rationality, typical for an economy based on the priority role of the price mechanism (Leal Filho et al. 2020). On the other hand, however, it causes some negative phenomena which, among other things, significantly limits the possibility of economically efficient long-term resource deployment. The second type of issue has its primary origin in the volatility of basic price parameters, which deprives economic operators of the possibility to determine the expected benefits

and costs reliably. The inability to determine the final effect of investment decisions prospectively, due to the lack of decision-making data, deprives investors of the primary instrument for assessing benefits and costs, i.e., economic account (Rabe et al. 2019). Not applying the above device of choosing the directions of capital allocation means consciously accepting the situation of implementation of inefficient mechanisms of using funds in the economy. Such a situation is mainly due to the occurrence of macroeconomic uncertainty (Di Marco et al. 2020). Therefore, taking the concept of sustainable development as a basis for my deliberations, I set the aim of the article to analyze the use of the economic calculus of an enterprise in conditions of higher risk. By adopting discounted methods of evaluating sustainable development investment projects as factors necessary for the rational economic behavior of an enterprise, I modified the Net Present Value (NPV) meter to take into account the variability of interest rates. In the first form, the algorithm assumes a change of the parameter "r" in each year(s) of the investment. In the second form, the algorithm takes into account the change of parameter "r" more often than once a year.

## 2. Literature Review

Modern company finance theory focuses its main interest on issues related to the behavior of business entities in the process of allocation and use of resources in spatial and temporal terms, as well as in various environmental conditions. This interest is closely related to the management process, within which we distinguish: (a) management goals; (b) resources, the number of which is always limited; (c) methods of the most appropriate use of resources to achieve the assumed goals. Problems of this kind are reflected in economic literature. Both the systemic aspects of management and the tools for achieving the set objectives are the subject of the research (Rajkovic 2020). The unique role of financial management as a factor influencing economic development is emphasized (Jones 2007; Barnett et al. 2015). In turn, shaping the financial economy in a company is determined by the construction of control systems for the entire economy. One of the primary structures of the economic system, choosing the role of financial management is the regulatory system. It manifests itself like tools for managing the economic system, which determines the independence of undertakings in terms of their decisions. Given the overriding role of money in shaping the economy, attention should be paid to the efficiency of the financial management process (Biddle and Hilary 2006; Gao and Yu 2018; Sameta and Jarbouib 2017). This is an essential factor in ensuring the achievement of the adopted objectives in the area of socio-economic policy and at the level of sustainable development. The theory of economic calculus is part of this type of issue.

There are many studies on the economic calculus of a company (Bosch and Newton 1995; Heal 1998; Donici and Încălțărău 2010; Joly 2016; Ziegler 2017; Baecker 2019; Boettke and Piano 2019). However, little is known about how to use variable interest rate economic calculus in the theory of sustainable development. I am trying to fill this research gap. In the assumed assumption, the uncertainty concerns in particular changes in the level of the interest rate, which is a primary indicator of the profitability of sustainable development investment. The limited ability to manage the risk of interest rate changes over long time horizons results in a material error in the objectivity and reliability of calculations of the economic efficiency of an investment.

## 3. Research Methodology

Observing contemporary trends in sustainable development, I have analyzed the rational process of capital allocation using economic calculation under conditions of risk and uncertainty. I then started a literature review to determine whether the economic calculus of sustainability is appropriate in unpredictable conditions? I am aware that literature study as the primary research method can be perceived as the limitations of research. However, I argue that literature research is a valuable source of evidence. Minor references in the literature on the subject to the risk and uncertainty in the economic calculus may indicate that such a solution may not have been taken into account. Consistent with

an abductive approach (Lukka 2014; Parker and Northcott 2016), the insights in this paper have emerged iteratively through consideration of both theory and the empirical cases.

The analysis is aimed at verifying the research hypothesis, which is the basis of the conducted considerations. The premise is as follows: taking a fixed discount rate reduces the objectivity (usefulness) of the results obtained when carrying out the economic account of sustainable development investment. In high-risk conditions, companies have to base their economic calculus on very volatile data. The volatility of the discount rate determines the level of the assumed targets (especially in the long term) that the entrepreneur adopts at the time of making a decision. It is known that the parameters of the economic calculus are subject to change as a result of the market mechanism. Still, their volatility is so high that it significantly hinders investment decisions. The above comments make it necessary to modify the algorithms containing the discount rate. The modification of the NPV algorithm presented in the article assumes different discount rates and their different duration.

## 4. Discussion and Results

There is only one economic calculus—an account based on data that provides a market mechanism that is sovereign about production. It is a characteristic of an enterprise in a market economy, which is geared to maximizing its monetary income, to act by the principle of sound management. However, in most cases, individual and organizational decision-makers choose satisfactory rather than optimal alternatives (Marengo 2020). From this point of view, taking into account the concepts of sustainable development, it can be concluded that economic calculus is a set of methods for measuring the inputs and outputs of business activities, aimed at providing management with information to facilitate rational and satisfactory investment decisions. Fourgeaud and Perrot (1990) consider that, to examine the rationality of a given investment project, the economic calculus should collect information on the investment project and assess the consequences that the implementation of the project would have on the overall variables considered. Given the isolated sustainability variables, the price parameters of the economic calculus, which take into account the high volatility of investment capital, should be taken into account.

On this basis, it should be noted that to be useful in the concept of sustainable development, the economic calculus in a company must meet the following assumptions:

1. The company keeps economic calculus. It is assumed to be an autonomous system (it has freedom of choice) and aims at survival or development. To this end, it maximizes value, as the only value can ensure the recovery of its resources in the next production cycle. It is also assumed that there is a qualitative and quantitative relationship between the company and its environment. It consists of the company producing for the environment with which it exchanges products. A business is an open system, which means that its growth and survival depends on the current and future external conditions. An available system is one that strives for a state of dynamic equilibrium and is capable of performing long-term work in the form of continuous exchange of matter and information with its environment. In such a system two opposite processes take place: the creation of negentropy, i.e., new configurations of elements from which the system is built, conditioning its efficient operation and the result of entropy—destruction during process of some aspects of the system, removed to the environment (Boehlke 2019). We can call today's business environment according to the theory of turbulent management. To operate in a highly turbulent environment, a company must continuously explore it (Mintzberg et al. 2009).
2. The subject of economic calculus is a process.
3. The production factors are divisive and can be used in the process with different intensity.
4. Effects and expenses are elements of the account. Standard units of measurement can be defined for these elements so that they can be aggregated (Boehlke 2020).

Taking into account the assumptions set out above, the economic calculus of the company can be determined:

1. The scope of the undertaking's independence should be as broad as factor substitutability. Any limitation on liberty is equivalent to a restriction of the economic calculus. From this point of view, it is possible to use economic calculus under the concept of sustainable development.
2. Factor substitution. The scope of the economic calculus is equivalent to that of substitution.
3. The degree to which the capital held can be divided.
4. Objective selection criteria set out by the principle of the recovery of funds.
5. The time horizon of operation.

Once the assumptions are made, the substance of economic calculus is to measure inputs and outputs with a time factor, to compare alternative combinations of results and information, and to choose modes of production. The problem of selecting a variety of production factors is formulated as follows: for each of the three elements (construction, technology, organization) the combination of production factors necessary for the manufacture of a given product can be determined. The amount of available inputs is limited. Each variety of production factors can be attributed to information—the effect of consumption of these factors and results expressed in monetary units. Efficiency functions (target function) can be defined as the relationship between outputs and inputs. Once these assumptions have been taken into account, the problem is to choose the three elements that would maximize the value of the organization's objective function at a given factor size.

The analysis of these assumptions leads to the identification of the company's economic calculus as an instrument of sustainable development, by selecting the most favorable option from the point of view of the adopted criteria, after preliminary multi-optional analysis of the available proposals. These definitions of economic calculus need to be complemented by taking into account factors that are currently of interest to the theory of sustainable development. This is particularly true of the time factor and the uncertainty (risk) that arises, among other things, from its nature. The following proposal should therefore supplement the definitions quoted above: an enterprise's economic calculus is an instrument for selecting the best option from among the considered, within a restrictive environment based on a specific selection criterion that takes into account the risk factors of future economic events. The economic calculus should therefore include consideration of problems related to the use of the directions of allocation of enterprise resources by the interest of sustainable development.

Using the economic calculus of an enterprise in the conditions of sustainable development is possible when:

- the effects and outlays incurred in connection with business activity are measurable (measurable),
- the impacts and outlays are expressed in the same units of measurement,
- the selection criterion is, as far as possible, clearly defined (Evans 2016).

The first rule is that a specific unit of measurement must be applied to the economic calculus. Apart from the (consumer and capital) market, it is money that is the most necessary condition for achieving maximum formal rationality. The financial system is characterized by formal rationality. They count on the ability of the system to meet the needs of its stakeholders. Tobias and Shin (2009) point out that a monetary calculation technique achieves the highest degree of formal rationality.

In contrast, many variables as possible gain monetary expression and, by this standard feature, can be the subject of comparisons—economic calculus. Not using money in economic calculus means that there is no possibility of substitution between different factors of production. After all, the financial calculus is about the various quality and not compatible with each other. Thus, the main weakness of the economic calculus in natural units remains the mismatch of its components.

The second condition seems to be logical in that it would be impossible to make any comparisons when expressing inputs and outputs in different units of measurement. Therefore, for the possibility of conducting an account, this postulate is considered logical

and natural. Application, for example, of the cash and quantity account would have to lead to the impossibility of making any rational calculations.

A third condition for the use of an economic account is that the selection criterion is relatively straightforward. The selection criterion is expressed in the form of a specific meter. The essence of the measures (used as selection criteria) is to provide information about economic events taking place in the company and its environment. There is a feedback between the meters and events, which consists in the fact that on the one hand, the meters provide information for new decisions, i.e., actions that cause events, and on the other hand—these recent actions cause changes in the meter. Thanks to this coupling, the meters can be a selection criterion, i.e., a function of the target.

Particular attention should be paid to the problem of risk when selecting the directions of capital allocation. The risk phenomenon is an integral part of the concept of sustainable economic development. Each capital allocation is characterized by a certain degree of risk (Rammel and van den Bergh 2003). In particular, the threat manifests itself in investment processes, where the scale of the lack of information is the result of a time gap between the moment of capital commitment and the possibility of its recovery.

It can be assumed that risk is an objective factor accompanying every entrepreneur who takes action. The differentiating factor for a socio-economic actor is the type of risk, the intensity of its impact and its consequences. Due to the uncertain situations in the management process, the threat becomes a factor that should be strongly emphasized in the theory of sustainable development. It can also be assumed that risk as an objective phenomenon, resulting from the operation of the market, is a factor introducing an element of rationality into the concept of sustainable development. As a consequence, the risk conditions force the entrepreneur to consider different decision-making options and to choose the most optimal and satisfactory option from the point of view of sustainable development objectives (Gramling and Schneider 2018; Klinke 2020).

The risk analysis of the concept of sustainable development, which is specific to sound business decisions and actions, should include a psychological element. From this point of view, the risks in the theory of sustainable development can be distinguished:

(a) a concept derived from Kningh (2013) work, which focuses on the entrepreneur's ability to bear risks,

(b) a trend related to the research carried out by Schumpeter (2003), which emphasizes the entrepreneur's ability to innovate and to be entrepreneurial,

(c) Fiedler's (1981) theory, which is based on the personality traits of the entrepreneur determined by situational conditions.

Each of these research concepts has a significant impact on the development of sustainability theories. It is worth noting the convergence between the risk aptitude approach and the set of human personality traits that determine the entrepreneur's behavior in specific situations (Le Blanc et al. 2020). In the functioning of an entrepreneur, we can distinguish two spheres: the sphere of determinations and the sphere of indeterminacy. The scope of each sphere depends on many situational factors. Particular attention in the concept of sustainable development should be paid to the globe of indefiniteness, the field of influence of which expands with the extension of the time horizon (Waas et al. 2011). In such a situation, investment decisions are associated with an increased level of uncertainty (Aven and Renn 2009). Delay is usually defined as the need for the decision-maker to decide on an activity without full information about the reality in which it will take place. From a statistical point of view, uncertainty refers to situations with the unknown probability distribution of future management conditions (Aven and Bouder 2020). About sustainable development, precarious situations are those whose outcome depends both on man and events beyond his control, but they cannot be predicted or measured.

Analyzing the effectiveness of investments under the account of sustainable development, one can quote the definition of risk proposed by Moyer et al. (2005). Researchers define risk as a possibility of differences between current and forecasted effects of an entrepreneur's actions. This raises two types of problems. Firstly, the risk increases the

difficulty in estimating future results. Secondly, according to Higgins (2009), risk on a conceptual level has a very significant impact on the value of investment projects. In general terms, it can be said that there is a risk of not achieving the objectives set for specific sustainable development decisions. Therefore, when assessing the economic effects of economic undertakings under sustainable development, it can be assumed that uncertainty and risk are a function of two factors:

- the quantity and quality of information at the disposal of the entrepreneur,
- the variability of conditions for the implementation and exploitation of the investment project.

In conclusion, it should be concluded that decisions of entrepreneurs pursuing sustainable development priorities should be made based on economic calculus, taking into account the diverse environment. It is worth recalling and further expanding the understanding of the definition of economic calculus. It has a broader scope of content, as its essence concerns a multidimensional approach to the problem of differentiated development. Economic calculus is a multifaceted combination of inputs and outputs, giving the choice of the optimal option of action from among the considered alternatives, based on the selection criterion, which is the economic efficiency of a given undertaking. In the concept of sustainable development, the financial calculus should be treated more as a way of thinking about the rationality of decisions taken by an entrepreneur. Planning in the long term is risky. In the case of the economic calculus of sustainable development, it is not just about obtaining accurate forecasts and their effects, but about choosing the direction of the desired action.

*Assessment of the Usefulness of Discounting Methods in the Economic Account of Sustainable Development Investments*

The financial management process is based on certain types of analytical techniques. These techniques are intended to allow the choice of an action option that is acceptable in terms of the objectives pursued (technical and financial feasibility). As part of the assessment of the economic efficiency of tangible investments in sustainable development, static, and dynamic methods can be distinguished (Raftery 2003). Consideration of the active processes of the economic calculus of the asset is determined by taking into account the concept of the value of money in time. The occurrence of inflationary factors, risks and uncertainties in the process of management makes taking such issues into account a reflection of rational action. The fact that the concept of the value of money in time is not used in the calculations of the economic efficiency of investment causes the expenses and effects to be equally valuable. This naturally contradicts the conditions of management, where it is known that money with today's "date" has a different value from money with a later "date".

The increase or decrease in the value of money is characteristic of both unstable economies and those characterized by macro and microeconomic stability. Changes in the value of money over time are most often identified with the impact of inflation factors (Thompson and Thompson 2020). However, it should be noted that an equally important factor influencing the value of money is the risk that accompanies the process of engaging funds. Thus, it can be assumed that the variable value of money is typical for all farming conditions. Of course, this variability will be less pronounced in areas where there are more stable conditions for doing business. Without going into a broad description of the reasons behind the existence of the variable value of money over time, it can be concluded that this is an essential issue for the valuation of economic events because it directly affects the objectivity and relevance of the calculus account calculations. This issue concerns, in particular, the economic calculus of sustainable development investments, where there is a long time horizon between the moment of incurring the expenditure and the moment of obtaining subsequent effects from the exploitation of the investment. This is a severe problem from the point of view of the effectiveness of the microeconomic sustainability mechanism.

Comparison of inputs and outputs, i.e., bringing them to the same level of time value, is possible using a discount technique (in the case of assessing the efficiency of tangible investments). This technique allows, by determining the value of the parameter "r" (discount rate), to choose the value of expected future effects at the moment of incurring the expenditure. The analysis of the investment literature allows us to conclude that the described dynamic methods of investment project assessment (NPV, Internal Rate of Return (IRR)) do not take into account the actual conditions of micro and macro-scale management (Militaru 2016; Nwogugu 2016). This is a severe problem from the point of view of improving the effectiveness of the microeconomic sustainability mechanism, as it is an essential source of information for businesses. It should be noted that one of the significant problems for the development of many economies in the world is the low level of investment in sustainable development, which is the driving force behind the development of the whole society (Sujit et al. 2020). This development is, therefore, primarily determined by the quality of financial management at the investor level (Stewart 1999).

The theoretical recommendations, referring to the broadly understood investment process, treat many issues in a simplified way (Chang et al. 2020). Much attention is paid to the theory of capital structure, trying to determine its optimal level from the point of view of maximizing the company's value (Ma et al. 2009). The issue of leverage is also being developed to increase the return on equity. The same applies to the estimation of equity and debt (Dierkes and Maeyer 2020). Many methods (from the simplest ones based on a risk-free rate to arbitration models) are indicated here. However, none of them develops the issue of dynamic recognition of the cost of capital by adopting statistics in the management process. This problem is one of the main factors limiting the reliability of the economic calculus, which is reflected in the assumption of a fixed discount rate over the life of the investment. This results in a situation where hypothetical invariability (staticity) of all factors that affect the determination of the cost of capital is assumed. This will distort the real conditions of management that can be observed, for example, with the level of base interest rates over time. This volatility occurs both in the short term (deviations from the previous group are both "in minus" and "in plus") and in a long time. Therefore, such a situation does not allow for an upbeat assessment of the usefulness of methods assuming interest (discount) rate fixity.

The above problem is the presentation of the Lombard rate and rediscount rate for bills of exchange in the period from January 2010 to November 2020 in Poland (Figure 1).

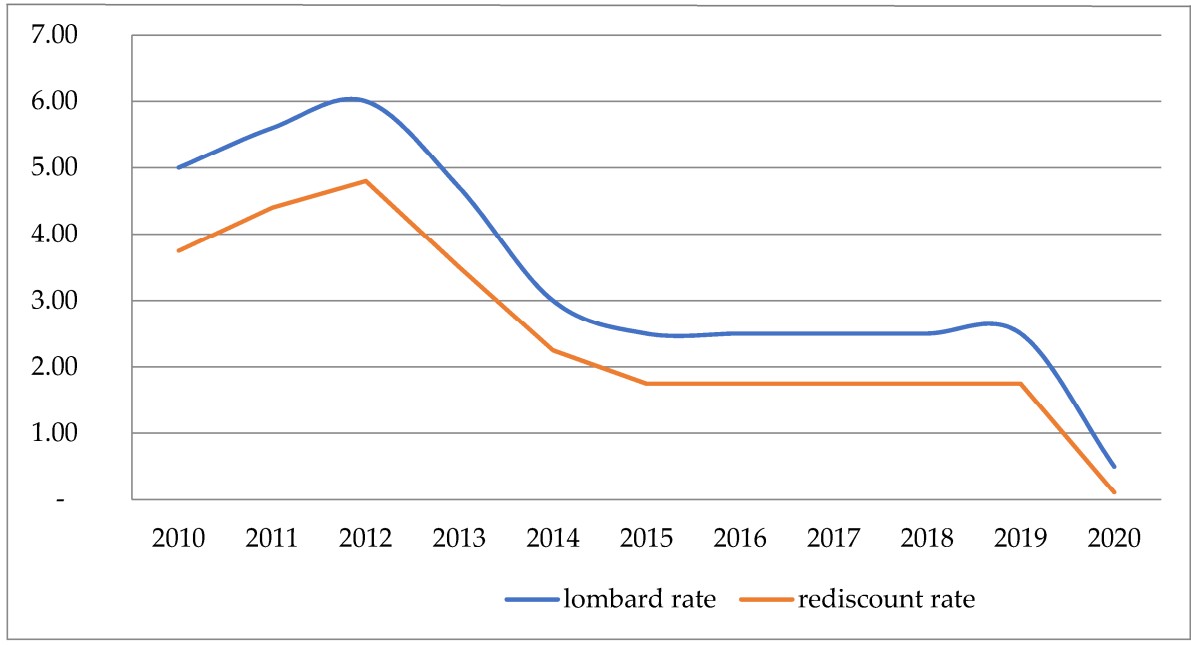

**Figure 1.** Lombard rate and rediscount level in Poland (2010–2020). Source: own study based on (NBP 2020).

The analysis of the above data confirms the observations made earlier about high-interest rate fluctuations. Within ten years, the interest rate (Lombard rate) has fallen from 5% to 0.5%. This means a difference of 4.5 percentage points. In the analyzed period, the pawn shopper's rate fluctuates "in plus" and "in minus". It can be noted that in May 2012, after a period of decline, the Lombard rate is increased (to 6.25%) by the monetary authorities and then lowered again. The situation is similar to the level of the rediscount rate for bills of exchange (see Figure 1)[1]. It should be noted that the rediscount rates together with WIBOR (Warsaw Interbank Offer Rate) are the interest rates to which the bank credit rates are indexed, which are the basis for calculating the cost of capital for the investment concerned.

Taking into account the construction of the meters that are the basis for investment decisions, I chose the NPV criterion for the research. This meter, together with its modified form, is as follows:

$$\overline{X} = NPV_F - NPV_V$$

where:

$\overline{X}$—deviation value,
$NPV_F$—net present value at a fixed discount rate,
$NPV_V$—net present value at a variable discount rate.

The modified form of the analyzed meter allows showing the size of the difference ($\overline{X}$) between the $NPV_F$ meter (assumes a constant discount rate) and the $NPV_V$ meter (takes actual discount rates). The development of the model, considering the discount rate changes once a year, is as follows:

$$X = \left( \frac{NCF_1}{(1+r)^1} + \frac{NCF_2}{(1+r)^2} + \ldots + \frac{NCF_n}{(1+r)^n} - I \right) - \left( \frac{NCF_1}{(1+r_1)} + \frac{NCF_2}{(1+r_1)(1+r_2)} + \ldots + \frac{NCF_n}{(1+r_1)(1+r_2) \ldots (1+r_n)} - I \right)$$

Assuming NPV > 0
Designations:

$\overline{X}$—deviation value,
$I$—investment outlays,
$NCF_1, NCF_2 \ldots NCF_n$—financial surpluses obtained in subsequent years of operation of the investment,
$n$—the next year of the calculation period (duration of the investment),
$r_1, r_2 \ldots r_n$—discount rates for each year (discount rates for specific years of the investment).

The assumption that the duration of parameter '$r$' is always one year is also a significant simplification which does not take account of actual farming conditions. If the discount rate is changed more than once a year, the present value multiplier shall take the following form (the algorithm allows to reflect real market conditions):

$$X = \left( \frac{NCF_1}{(1+r)^1} + \frac{NCF_2}{(1+r)^2} + \ldots + \frac{NCF_n}{(1+r)^n} - I \right)$$

$$- \left( \frac{NCF_1}{\left(1+\frac{r_{11}t_{11}}{360}\right)\left(1+\frac{r_{12}t_{12}}{360}\right) \wedge \left(1+\frac{r_{1m}t_{1m}}{360}\right)} + \frac{NCF_2}{\left(1+\frac{r_{11}t_{11}}{360}\right)\left(1+\frac{r_{12}t_{12}}{360}\right) \wedge \left(1+\frac{r_{1m}t_{1m}}{360}\right)\left(1+\frac{r_{21}t_{21}}{360}\right)\left(1+\frac{r_{22}t_{22}}{360}\right) \wedge \left(1+\frac{r_{2m}t_{2m}}{360}\right)} \right.$$

$$\left. + \frac{NCF_n}{1+\frac{r_{11}t_{11}}{360}\right)\left(1+\frac{r_{12}t_{12}}{360}\right) \wedge \left(1+\frac{r_{1m}t_{1m}}{360}\right)\left(1+\frac{r_{21}t_{21}}{360}\right)\left(1+\frac{r_{22}t_{22}}{360}\right) \wedge \left(1+\frac{r_{2m}t_{2m}}{360}\right) \wedge \left(1+\frac{r_{nm}t_{nm}}{360}\right)} - I \right)$$

where:

$r_{11} \ldots r_{1m}$—annual discount rates (1 to $m$) in the first calculation year,

---

1 For example, when comparing the interest rate levels in the stabilised UK economy in 2010 and 2020, we see differences of 0.25 percentage points (see Bank of England 2020).

$t_{11} \ldots t_{1m}$—the duration of the annual discount rates in the first calculation year, calculated in days (the sum of $t_{11}$ to $t_{1m}$ equal to 360 days),

$r_{21} \ldots r_{2m}$—annual discount rates (1 to $m$) in the second calculation year,

$t_{21} \ldots t_{2m}$—the duration of the annual discount rates in the second calculation year, calculated in days (the sum of $t_{21}$ to $t_{2m}$ equal to 360 days),

$r_{n1} \ldots r_{nm}$—annual discount rates (1 to $m$) in the calculation year $n$-th,

$t_{n1} \ldots t_{nm}$—the duration of the annual discount rates in the calculation year $n$-th (the sum of $t_{n1}$ to $t_{nm}$ equal to 360 days).

Modification of the NPV algorithm to a form assuming different discount rates and their different duration makes it possible to draw the following conclusions:

(a) An undertaking which makes its investment decision based on an NPV algorithm assuming a constant discount rate runs the risk that the calculations obtained are not objective.

(b) There is a risk of selecting the wrong design in case of a negative trend in interest rates.

(c) The permanent fluctuation of interest rates triggered by monetary policy makes it impossible to estimate their level for investors in the medium and long term, which leads to subjective assessment.

(d) There may be a negative interaction between the discount rates and the level of cash flows (an increase in the former and a decrease in the latter may affect the implementation of investment projects).

## 5. Conclusions

The economic calculus is not a panacea for solving all decision-making problems within the framework of sustainable development. It becomes necessary to take into account also other complementary methods, such as behavioral analysis. However, indeed the wrong way to solve decision making problems is and will be exclusively intuition. Intuitive, not supported by economic analysis, decision making may lead to a conclusion that is only by chance the most beneficial. Moreover, the presented risk issues accompanying the process of financial selection lead to the view that in many cases an entrepreneur making an investment decision on the allocation of capital is faced with a problem of uncertainty rather than risk. The observation of the economic environment of businesses, especially in the context of the current pandemic situation, leads to the conclusion that risk-based methods in the economic calculation are not sufficient in the management process. The high unpredictability of individual environmental parameters in the environment means that the risk of financial decisions remains very high. This undoubtedly hampers the selection process when using economic sustainability accounting.

In a competitive economy, economic calculus data are generally obtained from the market. The entrepreneur does not have all the necessary information to calculate the projected effects of his decisions. Taking into account the issues raised in the article concerning the risk occurring primarily in the assessment of economic efficiency of the investment, it seems that the way of solving the problem should take into account the area of macro-regulation, especially if the assumptions of the concept of sustainable development are taken into account. Assuming that companies make decisions based on external data over which they cannot influence, attention should be focused on the unfavorable environment for economic choices created by the operation of the market mechanism and a specific type of state monetary policy. The consequence of this assumption is to pay attention to exogenous factors influencing the resolution of allocation problems at the micro-level. Among the exogenous factors, a number of issues can be mentioned, but the institutional set-up of the economy has a decisive influence on the analysis of the problem. This arrangement is essential in creating a business framework for entrepreneurs. In this situation, each decision taken is a result of factors that result from the logic of the sustainable development system.

The collected empirical material and literature studies enable the following conclusions to be drawn, which may form the basis for further research investigations:

1. The use of dynamic methods based on the variable value of money over time in the assessment of the economic efficiency of an investment is the basis for an economically rational allocation of capital. Volatility reflected in the level of the discount rate has an impact on investment decisions in the field of sustainability.

2. The variable value of money over time influences investment decisions, particularly in an unstable environment (unstable economy), due to the inability to estimate the level of the discount rate over time. The frequency and extent of interest rate changes are essential.

3. The variable value of money over time affects the investment account more than the operator's performance. This is due to the time difference between the moment of incurring the effort and obtaining the effect.

4. With a high and variable discount rate even in short periods, the variable value of money over time has a significant impact on the objectivity of calculations.

5. In the assumptions of sustainable economic development, the issue of dynamic estimation of the effects of capital allocation should be described in more detail in the literature. Factors determining the level of calculated investment effects on the day of making decisions are often difficult to forecast. Thus, the issue of the volatility of conditions determining the achievement of sustainable development objectives at the time of making entrepreneurial decisions is burdened with high risk.

6. In the concept of sustainable development, a distinction should be made between the ex-ante economic calculus and the current economic calculus (management). The ex-ante economic calculus (it determines the directions and methods of investing) is a determinant of possible adaptation measures considered based on the current account.

7. Suppose we assume a high degree of divisibility and flexibility in sustainable development investments. In that case, the execution of the adjustment account is again faced with a lack of long-term prospective information.

8. The expected outcome of a decision should be objective. If some of the decision data is unreliable (burdened with too much uncertainty), we are dealing with a subjective determination that cannot be the basis for the decision.

9. Changes are a natural phenomenon in the economy, a phenomenon that threatens investors is the dynamics of changes that we are currently observing in connection with the COVID-19 pandemic.

10. A feature of a market economy is a risk. The use of an economic account is possible when the parameters of the investment calculation are probabilistic (the result of the decision depends not only on us, but also on external events that we think are influenced by, but we can estimate the probability of the expected result). Otherwise, the calculations are subjective inference, i.e., we are dealing with uncertainty.

11. The NPV concept used in practice is based on the idea of a flat profitability curve. Such an assumption can only be made for a stabilized economy.

12. The immature financial market in Poland does not offer instruments to hedge against the risk of changes in the discount rate over time. However, even stabilizing the underlying interest rates cannot be considered an entirely satisfactory solution. State involvement through the creation of guarantee funds becomes necessary. The State, based on the strategic objectives of individual industries in the field of sustainable development, should provide compensation for negative market variations in interest rates.

**Funding:** This research received no external funding.

**Institutional Review Board Statement:** Not applicable.

**Informed Consent Statement:** Not applicable.

**Data Availability Statement:** No new data were created or analyzed in this study. Data sharing is not applicable to this article.

**Conflicts of Interest:** The author declares no conflict of interest.

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
