# Peer review of "Economic Calculus Qua an Instrument to Support Sustainable Development under Increasing Risk"

_jrfm, doi:10.3390/jrfm14010015_

Round 1
Reviewer 1 Report
The proposed article is too general. It ”attempts to determine the importance of risk in economic calculus of the company as an instrument to support sustainable development”, but:
- it talks about companies in general, without any differentiation or specific analysis, like all the companies across the world would be acting and behaving the same;
- it tries to create a connection between risk and economic calculus, but without any data or specific info to show for it, just some general considerations;
- it only mentions generalities about risk and companies, and it hardly touches the sustainable development part;
- there is no proper risk analysis, just random information about it;
- in the end, there is no specific correlation between the aforementioned elements, and the conclusions just stipulate general statements instead of specific info.
Moreover, the "lit review" ”chapter” is far from complete, with only a few authors and works mentioned briefly, while the topic is touching 3 major topics that are covered by a huge amount of research: risks and risk impact in companies, economic calculus of a company and sustainable development.
The research methodology is 1 paragraph long and does not cover properly any of the needed elements regarding this topic. The research itself is not conducted correctly and not supported by data.
There is no specific data or case study presented or included.
There are a lot of statements that don't have the proper citation.
The article needs a major english revision.
Author Response
Dear Ms./Dear Mr.
Thank you for your suggestions and comments on my article. I have tried to take into account the proposed changes.
Here are my answers.
- I mean companies operating in an unstable economy, e. g. Poland and other countries where the change of discount rates has different frequency and scope (I modified the content of the revised article).
- I added to the article a model I created (point. 4. 1), and I have given in the chart (for example) the change in base rates applicable in Poland in the years (2010-2020).
- I supplemented the literature in the article with the issue of sustainable development. I stressed the impact of interest rate volatility on the implementation of investment projects within the framework of sustainable development.
- I performed the risk analysis in the NPV algorithm, and I modified (pt. 4. 1).
- I have clarified and improved the conclusions in the article summary.
- I expanded the description of the research methodology, extended the analysis of the literature on the subject, in the introduction of the article and in the article itself, I added new citations.
With best regards, Grzegorz Drozdowski
Reviewer 2 Report
Please link the literature review to the conclusions better.
Author Response
Dear Ms./Dear Mr.
In this article, I showed the practical application of the new NPV formula, which assumes variable discount rates in investment projects that can be implemented in the framework of sustainable development. This model carries the frequency and scope of changes in the discount rate. In the article, I also supplemented the literature and added and corrected conclusions.
With best regards, Grzegorz Drozdowski
Reviewer 3 Report
Review of Manuscript JRFM 1037171 Title: “Economic calculus qua an instrument to support sustainable development under increasing risk.”
The paper is purely theoretical, arguing for the importance of the assessment of risk in the economic calculus of the company as an instrument to support sustainable development. It addressing an interesting and important problem. Overall, the paper is well-written and the arguments seem reasonable. I do recall a working paper about ten years ago that made similar arguments and tried to implement an economic analysis of sustainability but ran into problems of converting non-economic outcomes into dollars. Unfortunately, I have been unable to cite that paper. It is not entirely clear from the current paper how to do this either, except the suggestion that the focus should be on risk. Sustainability is something the credit ratings agencies are already considering in their ratings analysis, albeit in a more subjective way.
I have a few comments, which I hope the author may be able to address.
Comments:
- Given the large number of interpretations of the meaning of the concept of sustainability in the literature (and its usual association with ESG goals), the paper requires a clear definition of ‘sustainable development’ upfront.
- I would argue that the concept of sustainable development is based on an analysis of the use of an enterprise's economic calculus in conditions of uncertainty more so than risk, where the cause and long-term effects of today’s business decisions are perhaps impossible to quantify accurately (more ‘unknowns’ than ‘knowns’). Hence, in the context of sustainability this would imply indeterminacy of economic analysis, resulting in decision-making becoming unavoidably subjective.
- The author states “In the case of the economic calculus of sustainable development, it is not just about obtaining accurate forecasts and their effects, but about choosing the direction of the desired action.” It is not entirely clear what is meant by “choosing the direction of the desired action”. Milton Friedman (1970) in his paper “The Social Responsibility of Business is to Increase its Profits” noted that “They [managers] are incredibly short-sighted and muddle-headed in matters that are outside their businesses, but which affect the possible survival of business in general.” He posed that the dilemma with the social responsibility of a business is the following: “If businessmen do have a social responsibility other than making maximum profits for stockholders, how are they to know what it is?”
- A major problem with current models of economic decision making is that they are largely based on discounted cash flow valuation models, which tend to exponentially discount events further in the future. Since in sustainability there is more weight on the future rather than the past, I wonder what type of economic model the author has in mind in terms of performing the calculations.
- Point 5, Conclusion “Regardless of the imperfections of the analytical methods, the main problem is the difficulty in collecting an adequate amount of information.” It is unclear what type of information is referred to here.
- Some of the citations in the text are missing from the list of references, e.g., Knight (2013).
Author Response
Dear Ms./Dear Mr.
Thank you for your suggestions and comments on my article. I have tried to take into account the proposed changes.
Here are my answers.
The revised article gained practical value. I proposed a modification of the NPV algorithm, which takes into account the volatility of interest rates. It is essential in long-term investments made by enterprises operating in an unstable economy. An example of such a country is, among others Poland and other Central and Eastern European countries, where the frequency and scope of changes in discount rates are significant. I presented it as an example of a period of 10 years (point 4.1). If the analysis period were even longer, the fluctuations in base rates would be even more significant. I have clarified and improved the conclusions in the article summary. I expanded the description of the research methodology, extended the analysis of the literature on the subject, in the introduction of the article and in the article itself, I added new citations. I added and changed the conclusions to the article based on the new thoughts.
With best regards, Grzegorz Drozdowski
Round 2
Reviewer 1 Report
The revised article covers now all the relevant topics needed, as well as proper citations. The clarification from the author, that the article is targeting only Poland companies, should be clearly stated in article's text as well.